# Effect of *Trichoderma asperellum* on Wheat Plants’ Biochemical and Molecular Responses, and Yield under Different Water Stress Conditions

**DOI:** 10.3390/ijms23126782

**Published:** 2022-06-17

**Authors:** María Illescas, María E. Morán-Diez, Ángel Emilio Martínez de Alba, Rosa Hermosa, Enrique Monte

**Affiliations:** Institute for Agribiotechnology Research (CIALE), Department of Microbiology and Genetics, University of Salamanca, Campus de Villamayor, C/Duero, 12, 37185 Salamanca, Spain; millesmor@usal.es (M.I.); me.morandiez@usal.es (M.E.M.-D.); aemarti@usal.es (Á.E.M.d.A.); emv@usal.es (E.M.)

**Keywords:** fungal phytohormones, IAA, ABA, CKs, ROS, antioxidant activity, dehydrins, proline, N genes, drought tolerance

## Abstract

Eight *Trichoderma* strains were evaluated for their potential to protect wheat seedlings against severe (no irrigation within two weeks) water stress (WS). Considering the plant fresh weight and phenotype, *T. asperellum* T140, which displays 1-aminocyclopropane-1-carboxylic acid deaminase activity and which is able to produce several phytohormones, was selected. The molecular and biochemical results obtained from 4-week-old wheat seedlings linked T140 application with a downregulation in the WS-response genes, a decrease in antioxidant activities, and a drop in the proline content, as well as low levels of hydrogen peroxide and malondialdehyde in response to severe WS. All of these responses are indicative of T140-primed seedlings having a higher tolerance to drought than those that are left untreated. A greenhouse assay performed under high nitrogen fertilization served to explore the long-term effects of T140 on wheat plants subjected to moderate (halved irrigation) WS. Even though all of the plants showed acclimation to moderate WS regardless of T140 application, there was a positive effect exerted by *T. asperellum* on the level of tolerance of the wheat plants to this stress. Strain T140 modulated the expression of a plant ABA-dependent WS marker and produced increased plant superoxide dismutase activity, which would explain the positive effect of *Trichoderma* on increasing crop yields under moderate WS conditions. The results demonstrate the effectiveness of *T. asperellum* T140 as a biostimulant for wheat plants under WS conditions, making them more tolerant to drought.

## 1. Introduction

Wheat is one of the world’s most widely grown crops, accounting for eight percent of global crop production, with 776 million metric tonnes being produced in 2020/21 [1]. However, these figures may not be sustained, as crop yields are expected to be negatively affected in the coming years as a result of climate change [2]. Drought is one of the most damaging consequences of climate change for crops, and wheat in particular is affected by water stress (WS), mainly during the reproductive phase, which has a negative impact on the production and grain quality, with yield reductions of up to 20% [3]. Thus, there is a need to improve drought tolerance in wheat in order to maintain yields under this currently unavoidable stress. Plants activate a large variety of mechanisms at the physiological, biochemical, and molecular levels, to deal with the harmful effect of WS. Reactive oxygen species (ROS), including hydrogen peroxide (H_2_O_2_), are produced in plants at basal levels as the result of aerobic metabolic processes; however, WS triggers ROS overproduction and accumulation in plant cells [4]. ROS cause extensive protein, DNA, and lipid damage, leading to lipid peroxidation, and eventually, cellular damage and death. Nevertheless, ROS also play a role as signaling molecules in response to stresses. Thus, plants have developed efficient ROS scavenging mechanisms in order to maintain the ROS balance [4,5,6,7]. Antioxidant enzymes such as superoxide dismutase (SOD), peroxidase (POD), and catalase (CAT) constitute the first line of defense in maintaining basal levels of ROS. Plants also respond to drought by modifying the expression of certain genes that can be considered as markers of this stress response, as is the case for those coding for the transcription factors NAC (no apical meristem (NAM), *Arabidopsis thaliana* transcription activation factor (ATAF1/2), and cup-shaped cotyledon (CUC2)), DREB (dehydration-responsive element binding), and dehydrins (DHN) [8]. Plant cells also produce osmolytes such as proline (Pro) and proteins such as DHN, which play osmoprotectant roles under oxidative stress conditions [7,9].

Nitrogen (N) is a plant essential nutrient that plays critical roles in key processes such as photosynthesis. In particular, the wheat crop yield and quality are fully dependent on N inputs [10,11]. In this sense, high rates of N fertilizers are applied by farmers to ensure competitive wheat yields. In spite of this, crop plants can only take up 40–50% of the N supplied, resulting in the significant addition of N to the environment, the consequence of which is a series of environmental and health impacts [12]. Therefore, it is necessary to rationalize the needs underlying N use by wheat crops to optimize the application of N-based fertilizers in the field. Plants take up N from the soil, mostly in the form of nitrate (NO_3_^−^) and ammonium (NH_4_^+^), by means of the corresponding nitrate (NRT) and ammonium (AMT) transporters [13]. Plant N assimilation is a complex process that is regulated by different enzymes, including the nitrate reductase (NIA), nitrite reductase (NiR), glutamine synthetase (GS), and glutamate synthase (GOGAT). However, the genes encoding such well-described transporters and enzymes in model plants are very numerous in wheat genomes, and their functions are not yet fully understood [14].

The use of plant-beneficial microorganisms as biofertilizers and bioprotectants is becoming increasingly popular for achieving high crop production with a low ecological impact [15]. In this regard, *Trichoderma* is a fungal genus that is widely distributed and that has an elevated biotechnological value [16]. Several *Trichoderma* species are currently used as biological control agents against phytopathogenic fungi, oomycetes, and nematodes [17]. Moreover, *Trichoderma* spp. can induce plant systemic responses to counteract biotic and abiotic stresses, as well as to promote growth [18,19]. Indeed, recent studies have demonstrated that *Trichoderma* application can enhance wheat tolerance to salinity [20,21], drought [6,22,23], and waterlogging [24]. However, it should not be forgotten that correct strain selection is necessary and is dependent on the interest being pursued [22]. It has been suggested that *Trichoderma’s* production of different phytohormones such as gibberellins (GA), cytokinins (CK), salicylic acid (SA), indole-3-acetic acid (IAA), and abscisic acid (ABA) contributes to balancing the plant phytohormone network and consequently to the signaling processes of the defense responses of plants to abiotic stresses [6,22,25,26]. Likewise, *Trichoderma* spp. can also modify the ethylene (ET) levels in the plant through 1-aminocyclopropane-1-carboxylic acid (ACC) deaminase (ACCD) activity, promoting growth [27], and enhancing the tolerance to abiotic stresses as a result [20,22,24].

*Trichoderma* can be useful for protecting plants from drought stress, but not all strains are able to put this ability into practice. The aim of this study was (i) to select a *Trichoderma* strain from a set of eight, representing different genotypes within the genus, to determine its ability to increase the tolerance of wheat plants to WS; (ii) to explore the production of phytohormones and ACCD activity related this ability in the selected strain (*T. asperellum* T140); (iii) to determine the extent to which T140 confers WS tolerance to wheat seedlings through the measurement of biochemical parameters and gene expression changes linked to plant responses to drought; and (iv) to validate the protective effect of T140 on wheat plants subjected to high N fertilization and moderate WS, and its role in maintaining grain yield and quality parameters at the greenhouse level.

## 2. Results

### 2.1. Trichoderma Strain Selection

Eight strains representative of the genetic diversity of *Trichoderma* genus (*T. parareesei* T6, *T. virens* T49, *T. longibrachiatum* T68, *T. spirale* T75, *T. koningii* T77, *T. harzianum* T115, *T. hamatum* T123, and *T. asperellum* T140) were included in a wheat seedling growth assay to determine whether any of them, when applied to seeds, conferred tolerance to severe WS conditions. The fresh and dry weights of the aerial parts obtained from 4-week-old wheat seedlings after being grown under optimal irrigation (OI) and WS (irrigation withdrawn during the third and fourth weeks) conditions are shown in Figure 1. Compared to their respective control plants, no differences in the fresh and dry weights were detected in *Trichoderma*-treated plants under OI. None of the *Trichoderma* strains improved the dry weight (DW) of the plants subjected to WS. However, under the WS conditions, the T140-treated plants showed a better phenotype and had a significantly higher fresh weight (FW) compared to the control plants (Figure 2 and Appendix A).

Since *Trichoderma* ACCD activity has been observed to protect plants from drought stress [20,22,24], this enzymatic activity was evaluated in the eight *Trichoderma* strains after growing them for four days in synthetic minimal medium (Appendix A). Strains T75 and T123 displayed the highest values (*p* < 0.05). Although strain T140 did not stand out for its ACCD activity, it was not the lowest either. As such, due to its better in planta behavior, the strain *T. asperellum* T140 was selected for further studies, including a greenhouse assay.

### 2.2. Phytohormone Production by Trichoderma Asperellum T140

As the biosynthesis of phytohormones in *Trichoderma* has been observed to help plants overcome WS [22,23,25], the production of eight phytohormones by strain T140 after cultivation in potato dextrose broth (PDB) with and without the addition of tryptophan (Trp) was measured. As PDB medium is composed of plant material, uninoculated media were used as a control. T140 production profiles of IAA, ABA, and the CKs dihydrozeatin (DHZ), isopentenyladenine (iP), and trans-zeatin (tZ) under the two Trp conditions are shown in Figure 3. GA1, GA4, and SA production was not detected under either of the two conditions. The biosynthesis of IAA, CKs iP, and tZ increased with the addition of Trp, unlike CK DHZ biosynthesis. ABA production did not seem to be affected by the presence of Trp in the medium. According to a two-way ANOVA, the combination of the factors “strain T140” and “culture medium” had an effect on the IAA, iP, DHZ, (*p* < 0.001), and tZ (*p* < 0.01) production.

### 2.3. Molecular and Biochemical Changes in Wheat Seedlings

Gene expression and biochemical parameters linked to WS plant responses have been analyzed in wheat seedlings to determine the role of strain T140 in the observed increase in tolerance to this stress.

The expression levels of four drought-response marker genes obtained using real-time quantitative PCR (qPCR) analysis in 4-week-old wheat seedlings, some of which were and some of which were not treated with T140, and subjected or not to WS (irrigation withdrawn during the third and fourth weeks), are shown in Figure 4. No gene expression differences were detected for *NAC2*, DREB2, DHN16, and *P5CR* between the control and T140-treated plants with OI. However, these genes were significantly up-regulated in the control plants subjected to WS. The T140-treated plants exposed to WS showed significantly lower expression levels for the marker genes considered, when compared to those of the stressed plants that were not treated with *Trichoderma*. A two-way ANOVA showed a significant effect on the expression of the four genes for the factors “irrigation condition” and “T140 application”. In terms of the combined effect of both factors, only *DREB2* was not significantly affected.

The effect of strain T140 on wheat seedlings subjected to WS was investigated via quantification of the levels of malondialdehyde (MDA), a product of lipid peroxidation, free Pro, and endogenous H_2_O_2_, as well as the antioxidant activities of SOD and POD (Figure 5). In the absence of WS, no differences were detected between the T140-treated and untreated plants, with the exception of POD activity, which decreased when T140 was applied. The highest levels of all of the parameters were recorded in the control plants under WS, and, with the exception of MDA, the application of T140 significantly decreased the studied parameters. Regardless of the existence of WS, the plants treated with T140 showed no significant differences in these parameters compared to the control conditions, with the exception of an increase in Pro levels. CAT activity was also measured (data not shown), but no significant differences were obtained for any of the treatments and conditions. The results of a two-way ANOVA showed that the variable “WS” affected the five studied parameters, and that the variable “T140 application” had an effect on the Pro content, and SOD and POD activity, while the Pro and H_2_O_2_ contents and SOD activity were modified by the combination of both variables.

### 2.4. Greenhouse Trial up to Grain Production

A long-term greenhouse assay was performed to evaluate the ability of *T. asperellum* T140 to enhance the WS (1/2 watering applied) tolerance of wheat plants grown under high basal N fertilization and subjected to two N top dressing (TD) applications. Growth and production traits, such as the DW of the aerial parts, grain yield per plant (GYP), grain number per plant (GNP), and spike number per plant (SNP) obtained from 28-week-old plants are shown in Table 1. No differences were observed in the DW, GYP, and GNP between the untreated and T140-treated plants under OI, but the T140-treated plants showed significantly higher values of DW and GNP than the control plants under the WS conditions. Close GYP values were obtained between the untreated and T140-treated plants under OI. In the plants subjected to WS, there was a 20% GYP increase in those treated with T140. The two-way ANOVA results showed that WS led to a significant reduction in the DW (*p* < 0.001), GYP, GNP, and SNP (*p* < 0.01); the application of T140 affected SNP alone (*p* < 0.05), and no effects caused by the combination of both variables were observed. In addition, the grain yield (GY) and spike number (SN) values increased by approximately 20% in the T140-treated plants subjected to WS compared to the WS control plants.

In order to determinate the associations between grain quality and T140 application and irrigation conditions, the contents of seven macro- and six microelements were quantified in the wheat grain samples that were harvested in the greenhouse assay. All of the data and statistical analyses are shown in Appendix A. The significance of applying T140, WS, or both is shown in Figure 6. T140 application affected grain quality by increasing the Ca, Mg, P (*p* < 0.05), S, Cu (*p* < 0.01), and Fe (*p* < 0.001) contents, and reducing the B (*p* < 0.001) content. WS affected the contents of most of the macro- and microelements that were tested, reducing C (*p* < 0.01), Ca, and B (*p* < 0.001) and increasing the contents of N, K (*p* < 0.05), S, Mn, Zn (*p* < 0.01), and Cu (*p* < 0.001). The combined application of T140 and WS only significantly increased the C content (*p* < 0.01).

The qPCR expression levels of the eight genes involved in the drought responses or N metabolism in 25-week-old wheat plants are shown in Figure 7. No differences in the expression of *DREB2*, *DHN16,* and *P5CR* were observed between OI and the WS control plants, while the *NAC2* levels increased in the control plants subjected to WS (Figure 7A–D). Differences in the expression of the ammonium transporter 3 (*AMT3.1*) gene were only detected between the WS control plants and OI plants treated with T140 (Figure 7E). The T140-treated plants showed higher expression levels of the pleiotropic drug resistance 1 (*PDR1*) gene than the control plants, and, in turn, they were higher for the T140-treated plants with OI than they were for those under WS (Figure 7F). WS decreased the expression of both the nitrate reductase 7 (*NIA7*) and glutamine synthase 2 (*GS2*) genes in the control plants. However, only the T140-treated plants that were subjected to WS showed lower levels of *NIA7* compared to those with OI (Figure 7G,H). A two-way ANOVA showed that “T140 application” affected the expression of *NAC2*, *AMT3.1* (*p* < 0.05), *NIA7* (*p* < 0.01), and *PDR1* (*p* < 0.001), while “WS” modified that of *NAC2*, *DREB2*, *GS2* (*p* < 0.05), and *NIA7* (*p* < 0.01), and only *NAC2* and *PDR1* (*p* < 0.05) were affected by the combination of both variables.

The MDA and endogenous H_2_O_2_ contents, and the antioxidant enzyme activities of SOD, POD, and CAT quantified in 25-week-old wheat plants from the greenhouse assay are shown in Figure 8. A significant MDA increase was observed in the WS plants compared to those with OI, irrespective of whether T140 was applied or not (Figure 8A). However, the endogenous H_2_O_2_ content did not show significant differences between OI and WS plants, nor in the plants that did or did not receive T140 application (Figure 8B). The WS control plants displayed a significant decrease in SOD activity. Nevertheless, no differences were observed in the POD and CAT activity in the plants from any of the treatments and irrigation conditions. The results derived from a two-way ANOVA indicated that the MDA content (*p* < 0.01) and SOD activity (*p* < 0.05) were affected by the factor “WS”; the combination of “T140 application” and “WS” only affected SOD activity (*p* < 0.01).

## 3. Discussion

### 3.1. Trichoderma Strain Selection

It is well known that *Trichoderma* increases the tolerance of wheat plants to abiotic stresses such as salinity, waterlogging, and drought [6,20,24], although this trait is not an attribute of all strains of the genus [22]. Considering the FW of 4-week-old seedlings subjected to non-irrigation conditions within the previous two weeks and their phenotype, *T. asperellum* T140 was selected for further studies. As observed by other authors [7], differences were observed in the FW values between the plants with OI and WS, but this was not the case for the DW of the aerial parts. Our results obtained with strain T140 are indicative of an absence of wheat plant growth promotion, regardless of WS. Previous work linked the ACCD activity of *Trichoderma* to the ability of the producing strains to alleviate wheat plants from different abiotic stresses [6,20,22,24]. However, strain T140 did not stand out as having significant ACCD activity compared to the other *Trichoderma* strains that were tested, with the exception of strain T6.

### 3.2. Phytohormone Production by T. asperellum T140

Numerous investigations have shown that the phytohormones produced by root-associated microbes have been proven to be important for inducing host tolerance to abiotic stresses [28,29,30]. We detected the production of IAA, ABA, and CKs but not the production of SA, GA_1_, and GA_4_ in the T140 strain, a feature that depends on the culture medium and the *Trichoderma* strain that was tested [6,22,26]. Although ABA and CKs have antagonistic roles in regulating water use by the plant [31], it has been observed that exogenous supplies of one or the other phytohormone increase the drought tolerance of wheat plants [32,33]. The production profile of IAA and ABA by *Trichoderma* strains has also been related to their ability to help plants tolerate abiotic stresses [28,34], including wheat plants [6,20,22]. In this regard, strain T140 produced more IAA and ABA than *T. simmonsii* T137, a strain that was previously selected for its ability to confer drought tolerance to wheat plants [22]. The plant phytohormone balance plays a pivotal role in the physiological, cellular, and molecular processes that govern tolerance to abiotic stresses [29]. The IAA-, GA- and ABA-producing strain *T. asperellum* Q1 stimulated the FW biomass of cucumber seedlings under salt stress compared to the untreated control plants, while increasing the concentrations of these phytohormones in cucumber leaves [28]. Similar observations have been reported in other plants for plant growth-promoting bacteria [30]. This means that not only the phytohormones produced by *Trichoderma* but also the modification of plant phytohormone profiles in response to *Trichoderma* colonization condition plant responses to abiotic stresses [35].

### 3.3. Molecular and Biochemical Changes in Wheat Seedlings

In order to understand the role of strain T140 in the observed WS tolerance of wheat seedlings, parameters related to plant responses to drought stress were analyzed [36]. As expected, the absence of irrigation for two weeks elevated the expression levels of both the ABA-dependent (*NAC2*, *DHN16*, and *P5CR*) and ABA-independent (*DREB2*) drought response marker genes in control plants. *DHN16* encodes a DHN protein, and DHN proteins are induced by ABA and exhibit ROS scavenging properties [36], while *P5CR* is involved in the biosynthesis of Pro, which acts as an osmolyte that is used by plants to cope with drought stress [37]. Although no differences in the expression of these genes were observed between the T140-treated and non-treated seedlings with OI, under WS, those challenged with T140 always showed significantly lower expression levels than their respective controls. We observed that control seedlings subjected to WS had the highest levels of H_2_O_2_ and MDA, as well as SOD and POD, with the highest antioxidant activity and Pro content. Regarding Pro levels, the control plants under WS conditions showed a 350-fold increase compared to those with OI. As previously described [36], the osmolyte production and antioxidant machinery of the plant are not sufficient to protect it from severe drought stress, and in view of the damaged phenotype of the control seedlings of our study, this seems to have occurred. The lowering of the MDA and Pro levels detected in the T140-treated seedlings in response to WS compared to those from the untreated agrees with that described in wheat plants protected by *T. harzianum* against WS [38]. This is not a unique response attributable to *Trichoderma,* as mycorrhizal plants also decrease Pro levels when subjected to WS [39]. Previous studies [6,22] have reported that some *Trichoderma* strains exhibit a protective capacity for wheat plants against drought stress via the activation of the plant’s antioxidant enzyme machinery. In the present study, the absence of differences for SOD and POD activities between OI and WS T140-treated seedlings is in line with the similar H_2_O_2_ levels that were detected. Overall, the results are consistent with the phenotype observed in the T140 selection assay, and they may indicate that seedlings treated with this strain tolerated the level of applied WS much better than those that were untreated.

### 3.4. Greenhouse Trial up to Grain Production

Wheat is a crop that is highly dependent on chemical fertilization, and we conducted a greenhouse trial by applying an amount of inorganic N fertilization that is similar to that commonly used in extensive wheat production [11]. In this assay, two irrigation regimes were included to explore the effect of strain T140 on the WS tolerance of wheat plants in terms of yield. WS was performed by halving irrigation throughout the trial and obtaining a 30% reduction in the field capacity, representing a moderate stress condition, but not one that was drastic enough to bring the plants to final production. Studies in wheat have shown that WS decreases N use efficiency and crop productivity [40,41,42]. It has also been reported that the effect of *Trichoderma* on wheat plants subjected to moderate WS depends on N fertilization levels, and that only growth and biomass were increased under high N inputs [43]. As expected, the WS applied in our study lowered DW, GYP, GNP, and SNP regardless of T140 application. Although strain T140 did not appear to improve the parameters related to plant production under OI, wheat seed biopriming with T140 was significantly beneficial for WS plants in terms of DW and GNP. Previous studies have reported that *Trichoderma* strains that are capable of alleviating drought stress in wheat plants can also act as growth promoters in the absence of stress [22,43,44]. However, the opposite has also been described [6], supporting the idea that the *Trichoderma* biostimulation properties under different plant growth conditions is a strain-dependent trait. Our yield results have shown that under moderate WS, the application of T140 increased the percentages of GY (19.5%), GN (19.8%), and SN (18.1%). *Trichoderma* application also conditions the macro- and microelement levels in wheat plants [43] and in the subsequently harvested grains [11]. In the present study, the effect of T140 on the element content in the grains was smaller to that exerted by the WS and was not appreciable, with the exception of C, when T140 application and WS were combined. Previous studies have reported that drought stress improved the grain protein content [45,46]. It is not easy to draw conclusions, when T140 application, WS, and T140 application × WS positively affected the levels of P, N, and C, respectively.

To explore the plant mechanisms affected by T140 application under WS that could be involved in the improvement of growth, and the agronomic parameters observed in the greenhouse assay, we analyzed gene markers, antioxidant activities, and drought-related metabolites. WS did not significantly modify the expression levels of the two transporters, *AMT-3.1* and *PDR1*, but it reduced those of the two N metabolism genes, *GS2* and *NIA7*. Controversial results on the expression of ammonium transporter genes in WS wheat plants have been reported [41,47]. While a marked induction of ammonium transporters by WS has been described in the vegetative stage [47], plants in later developmental stages did not show upregulation under WS, but they did so under N deficit [41]. According to this mentioned study [41] and considering that we have been working with high N fertilization doses, we did not observe changes in the expression of the ammonium transporter *AMT3.1* in 25-week-old plants, which could be associated neither with WS nor with T140 application. We also were unable to determine whether WS affected *PDR1* gene expression. However, the *PDR1* levels were modified via the application of *Trichoderma* and the combined action of T140 and WS. Plant PDR ATP-binding cassette (ABC) transporters family proteins play an important role in the detoxification process by preventing water loss [48]. These ABC transporter genes are up-regulated in wheat plants when stressed by N starvation [49]. The upregulation that we observed seems to be more related to the contribution of strain T140, as the plants were grown under a non-limiting N supply, as observed in a previous transcriptomic study conducted on wheat seedlings [50]. The detected downregulation of the N-metabolism genes under WS conditions is in agreement with the described decrease in *GS2* expression in wheat plants [51] and NIA activity in sugarcane [23] caused by drought. In this sense, we also observed that while *GS2* expression was not affected by T140 application, *NIA7* expression increased due to T140 application in the two tested irrigation conditions compared to their respective controls, although it was always lower when the plants were subjected to WS. The observed *NIA7* expression profiles agree with the NIA activities measured on non-stressed and WS sugarcane plants challenged with *T. asperellum* [23]. A recent study identified nine *NIA* genes in the wheat genome and showed their involvement in wheat–microbe interactions [52]. Work in Arabidopsis has reported that the regulation of *NIA* genes is important for plant development and growth, and some of them play a key role in abiotic stress adaptation [53]. In our greenhouse assay, nitrate was not applied since we used NPK as basal and Nitrosulfam (NS) as TD N fertilization. It has been reported that in addition to nitrate, many factors can also affect the expression of the NIA genes, ABA signaling, ET signaling, and N metabolites (mainly glutamine), causing them to function as negative regulators [53].

We also explored *NAC2*, *DREB2*, *DHN16,* and *P5CR* expression levels in the greenhouse-grown plants, and although the *DHN16* and *P5CR* levels were not affected by the two factors that were considered or their combination, *NAC2* was up-regulated by T140 application and WS, but not by their combination. In contrast, the *DREB2* levels appeared to be unaffected by the application of T140 or the combination of T140 and WS, this being in contradiction with what was reported for the *Trichoderma*-treated tomato plants subjected to saline stress [35]. A decline in plant growth accompanied by increases in lipid peroxidation (MDA levels), antioxidant enzyme activities, and H_2_O_2_ and osmolyte accumulation could be expected in WS plants [5,7,54]. In our study, WS led to similar MDA levels in both the control and T140-treated plants, and no changes between them were observed in the H_2_O_2_ and osmolyte levels. Doubling the MDA content in drought-stressed wheat plants has also been reported in previous studies [5,7]. However, the differences in the MDA levels recorded between our plants and those of these two previous studies [5,7] may be due to the degree of the WS applied and its duration, the age of the plants, and the methodology used to calculate them. The marked differences with what was observed in wheat seedlings subjected to severe WS regarding to the expression of marker genes, antioxidant enzymatic activities, and H_2_O_2_ and osmolyte contents, may also be partly due to differences in the form of WS applied. In addition, comparisons considering the differences between both assays could not be made. As the plants treated with T140 and subjected to WS showed higher biomass and yield parameters, accompanied by higher SOD activity than the WS control plants, it can be deduced that this *Trichoderma* strain exerts a protective effect on wheat against drought.

## 4. Materials and Methods

### 4.1. Trichoderma Strains

Eight *Trichoderma* strains, representing different genotypes, were included in this study: *T. parareesei* T6 (air, UK) [55,56], *T. virens* T49 (soil, Brazil), *T. longibrachiatum* T68 (soil, Brazil), *T. spirale* T75 (solarized soil, Spain), *T. koningii* T77 (soil, Spain) [6,55], and *T. harzianum* T115 (soil, Philippines) [6], *T. hamatum* T123 (*Marchantia polymorpha* rhizoids, Spain), and *T. asperellum* T140 (strawberry nursery soil, Spain) (references of our collection, CIALE, University of Salamanca, Spain). Strains were routinely grown on potato dextrose agar (PDA, Difco Laboratories, Detroit, MI, USA) at 28 °C in the dark. For long-term storage, the strains were maintained at −80 °C in a 30% glycerol solution. Conidia from 7-day-old PDA plates were harvested, and the conidia concentrations were calculated as previously described [35].

### 4.2. 1-aminocyclopropane-1-carboxylate Deaminase Activity of Trichoderma Strains

The ACCD activities of the T6, T49, T68, T75, T77, T115, T123, and T140 strains were measured as previously described [6]. For each strain, 10 mL of synthetic medium [57] was inoculated with 100 µL of conidial suspension (1 × 10^6^ conidia/mL), and the cultures were grown at 180 rpm and at 28 °C for 4 days. The mycelia were collected and homogenized in 2.5 mL Tris buffer 0.1 M (pH 8.5). ACCD activity was determined in a colorimetric assay via the measurement of the amount of α-ketobutyrate produced by the deamination of ACC at 540 nm. A standard curve, prepared with α-ketobutyrate (10–1000 µmol), was used as a reference. ACCD activity was expressed as mmol of α-ketobutyrate formed in 1 h, and specific ACCD activity was expressed per mg of protein. Quantitative protein determination was performed using a Bradford assay [58]. Three biological replicates were analyzed for each strain, and activity measurements were also performed in triplicate.

### 4.3. Determination of Phytohormone-like Compounds in Trichoderma

The strain *T. asperellum* T140 was grown in 200 mL of PDB and PDB with 200 mg/L of tryptophan (PDB-Trp) media as previously described [6], and the culture supernatants were collected via filtration. In parallel, uninoculated PDB and PDB-Trp media were used as controls. The supernatants were lyophilized, the DW was measured, and they were kept at 4 °C until hormone extraction. A 50 mg (DW) amount of lyophilized supernatants was used for GA, IAA, ABA, SA, and CK quantification using a previously described methodology [6]. Results are expressed for IAA in μmoles and for the other phytohormones in pmoles per liter of culture supernatant. Three independent replicates were analyzed for each culture medium.

### 4.4. Trichoderma Assays on Wheat Seedlings under Different Irrigation Conditions

Wheat seeds (*Triticum aestivum* L., variety Berdun) were surface disinfected and stratified as previously described [6]. *Trichoderma* was applied to the seeds through coating them with 2 × 10^6^ conidia/seed, as previously described [11]. Seeds were sown in 50 mL tubes (two seeds per tube) containing a vermiculite/perlite (1:1) mixture that had been previously autoclaved at 121 °C.

A first assay included the nine following treatments: control (*Trichoderma*-untreated seeds), T6, T49, T68, T75, T77, T115, T123, and T140. Seedlings were maintained in a growth chamber under 60% humidity, 22 °C, and a 16 h light/8 h dark photoperiod, and they were watered twice a week with 2 mL of water and supplemented with 1.2 mL Hoagland solution [59] per tube once a week. Two weeks after sowing, 36 seedlings from each treatment group were separated in two blocks: (i) 18 plants were maintained with OI, and (ii) 18 plants were not watered for 2 weeks (severe WS). This experiment lasted 4 weeks. The plants were photographed, and their aerial parts were collected and weighted.

A second assay was carried out as described above, that included just two treatments: control (*Trichoderma*-untreated seeds) and T140. The aerial parts of 4-week-old wheat seedlings from both treatments and irrigation conditions were collected in three sets of three plants for biochemical and qPCR analyses.

### 4.5. Greenhouse Assay with Trichoderma and Wheat Plants under Different Irrigation Conditions

Surface-disinfected wheat seeds were sown in 2.5 L pots (three seeds per pot) containing a sterile mixture of commercial peat (Projar Professional, Comercial Projar SA, Fuente de Saz, Madrid, Spain) and vermiculite (3:1). This assay included two treatments (T140 and *Trichoderma*-untreated seeds as control) and two irrigation conditions (OI and WS). The wheat seeds were coated with a T140-conidial suspension, as indicated above, which was replaced by sterile distilled water to coat the seeds used as controls. A 240 mg amount of NPK 8-16-8 fertilizer was applied as the basal N fertilization in each pot. In addition, two TD applications of Nitrosulfam 46 25-0-0 MgO (NS) fertilizer (Mirat Fertilizantes, Salamanca, Spain) were made as follows: the first TD was performed 5 weeks after sowing (leaf development stage) with the 60% N requirements (218 mg NS per pot), and the second TD was supplied 13 weeks after sowing (tillering stage) with the 40% N requirements (145 mg NS per pot). Plants were maintained in a greenhouse that was temperature controlled to 22 ± 4 °C, as previously described [35], and watered as needed for 4 weeks. At this point, 72 plants from each of the treatments described above were separated into two blocks: (i) 36 plants were maintained with OI (90% field capacity); and (ii) 36 plants were subjected to WS via the application of 1/2 irrigation (60% field capacity) until the end of the assay (moderate WS). The experiment lasted 7 months. One leaf from the top of the stems of the 25-week-old wheat plants was collected, immediately frozen in liquid nitrogen, and ground for further biochemical and qPCR analyses. Sampling was also conducted at the end of the cultivation period, 28 weeks after sowing, to measure the aerial DW and GYP and to count the GNP and SNP. In addition, the GY and SN as well as macro- and microelement contents in the grains were determined.

### 4.6. Chemical Properties of Wheat Grains

The wheat grain contents of C and N, and the macro- and microelements were quantified by IRNASA’s analytical service (CSIC, Salamanca, Spain). Five biological replicates per treatment and condition were analyzed. The contents of macro- (S, P, Mg, K, and Ca) and microelements (Fe, Mn, Zn, and Cu) were determined as previously described [11].

### 4.7. Biochemical Analyses of Plants

#### 4.7.1. Determination of Antioxidant Enzymatic Activities

Proteins were extracted through homogenizing 50 mg of frozen plant material in 1 mL of 50 mM potassium phosphate buffer (pH 7.8) and centrifugating the material at 10,000× *g* for 20 min at 4 °C; later, the supernatant was taken and used to estimate the antioxidant activities of SOD, POD, and CAT, as previously described [6]. The protein concentration was measured using a Bradford assay using Dye Reagent Concentrate (Bio-Rad Laboratories GmbH, München, Germany) and bovine serum albumin as a protein standard. The specific activity of CAT, POD, and SOD was expressed as units per min per mg protein. Three biological replicates per treatment and condition were analyzed.

#### 4.7.2. Determination of H_2_O_2_ Content

H_2_O_2_ quantification was assayed in 50 mg of frozen plant material, as previously described [60]. The absorbance of the supernatant was determined in a spectrophotometer at 390 nm, and the H_2_O_2_ content was extrapolated from a standard curve. Three biological replicates per treatment and condition were analyzed.

#### 4.7.3. Lipid Peroxidation

The MDA concentration was determined as previously described [7,61], with minor modifications. A 50 mg amount of frozen plant material was homogenized in 1 mL of 10% trichloroacetic acid and centrifuged at 8000× *g* for 10 min. A 1 mL amount of 10% trichloroacetic acid containing 0.6% thiobarbituric acid was added to 300 µL of the supernatant. The mixture was heated at 95 °C for 30 min and then quickly cooled in an ice bath. After centrifugation at 8000× *g* for 10 min, the absorbance of the supernatant was determined in a spectrophotometer at 532 and 600 nm. The MDA content was calculated using 155 mM^−1^ cm^−1^ as the extinction molar coefficient [7]. Three biological replicates per treatment and condition were analyzed.

#### 4.7.4. Determination of Free Proline Content

The free Pro content was measured as previously described [62], with minor modifications. A 50 mg amount of frozen plant material was homogenized in 750 µL of 3% sulphosalicylic acid, and the residue was removed via centrifugation. A 100 μL amount of the extract was mixed with 2 mL of glacial acetic acid and 2 mL acid ninhydrin (1.25 g ninhydrin warmed in 30 mL glacial acetic acid and 20 mL of 6 M phosphoric acid until dissolved) for 1 h at 100 °C, and the reaction was then cooled in an ice bath. The reaction mixture was extracted with 1 mL of toluene. The chromophore-containing toluene was warmed to room temperature, and its absorbance was measured at 520 nm. The amount of Pro was extrapolated to be in the range of 20–100 µg, from a standard curve. Three biological replicates per treatment and condition were analyzed.

### 4.8. Real-Time Quantitative PCR (qPCR)

Expression analyses were performed with wheat leaves sampled from the two in planta assays described above. The total RNA was extracted from the frozen plant material using TRIZOL reagent (Invitrogen Life Technologies, Carlsbad, CA, USA), following the manufacturer’s instructions. The cDNA was synthesized from 1 μg of RNA, which was treated with DNase RQ1 (Promega Biotech Ibérica, Alcobendas, Spain) and then used for reverse transcription with an oligo(dT) primer using a Transcriptor First Strand cDNA Synthesis Kit (Takara, Inc., Tokyo, Japan) following the manufacturer’s protocols. PCR reactions were performed on a StepOnePlus thermocycler (Applied Biosystems, Foster City, CA, USA) and using SYBR FAST KAPA qPCR (Biosystems, Buenos Aires, Argentina). The reaction mixtures and amplification conditions were as previously described [35]. Each reaction was conducted in a total volume of 10 μL, and three technical replicates were used for each treatment and condition. The expression levels of *DREB2*, *NAC2*, *DHN16*, *P5CR, AMT3.1*, *PDR1*, *NIA7*, and *GS2* genes were analyzed. Primer sequences, both those that were previous described [50,63] or designed in this study, are listed in Appendix A. The Ct values were normalized with the values of the wheat ubiquitin gene, and the relative gene expression was calculated using the 2^−ΔΔCT^ method [64].

### 4.9. Statistical Analysis

All data were collected from at least three biological replicates. The homogeneity of variance and normality tests were performed using the Levene and Shapiro–Wilk tests. IBM SPSS Statistics 27 (IBM Corp., Armonk, NY, USA) was used for the statistical analyses through an analysis of variance (ANOVA) to identify significant differences among treatments, followed by a mean separation using Tukey’s or Duncan’s (for gene expression) test (*p* < 0.05).

## 5. Conclusions

The improved drought tolerance of wheat by *Trichoderma* remains a promising and challenging solution; however, the ways in which plants respond to this stimulus are not fully understood, and to a large extent, its implementation depends on correct strain selection. Since not all *Trichoderma* strains protect plants to the same extent from drought stress, careful selection, including plant trials and the measurement of biochemical and molecular parameters in both the fungus and the plant, is needed. *T. asperellum* T140 was the most promising strain out of eight strains that were tested. Molecular and biochemical data from T140-primed seedlings indicate that they suffered less from the severe WS applied compared to those that were left untreated, which was supported by the downregulation of the WS response genes, the low antioxidant machinery, and a drop in the Pro, H_2_O_2_, and MDA contents. However, our greenhouse assay performed under high N fertilization conditions seems to indicate that the wheat plants became acclimated to moderate WS, regardless of T140 application. The qPCR results showed that strain T140 significantly affected the expression levels of *NAC2*, an ABA-dependent WS marker, and those of the genes involved in N uptake and metabolism, such as *AMT3.1* and *NIA7*, respectively. Moreover, *NAC2* expression and SOD activity were particularly affected by the combination of T140 application and WS. These results, as well as an increased crop yield linked to T140 application, would be indicative of the role of *Trichoderma* in acclimatizing plants under stressful conditions.

## Figures and Tables

**Figure 1 ijms-23-06782-f001:**
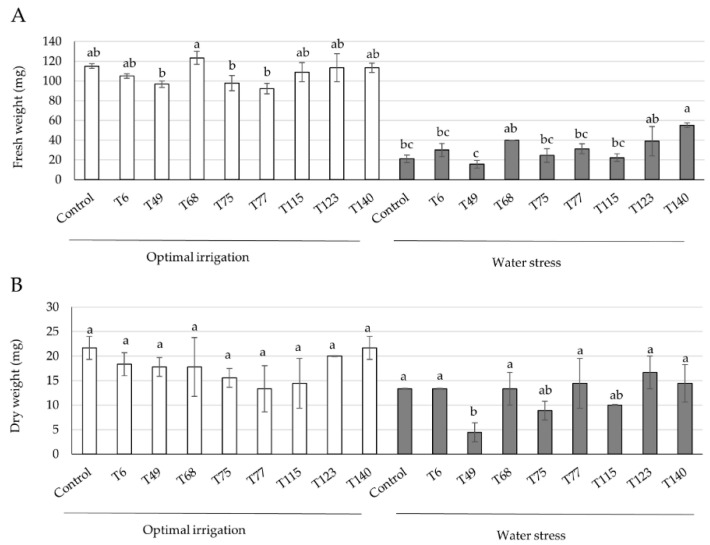
Fresh (**A**) and dry (**B**) weights of the aerial parts of 4-week-old wheat seedlings derived from *Trichoderma* (T6-T140)-treated and -untreated (control) seeds grown under optimal irrigation and water stress (irrigation withdrawn during the third and fourth weeks) conditions. The values are the means of three biological replicates per treatment and condition (*n* = 3), and the corresponding standard deviations. For each condition, different letters above the bars indicate significant differences according to one-way analysis of variance (ANOVA) followed by Tukey’s test (*p* < 0.05).

**Figure 2 ijms-23-06782-f002:**
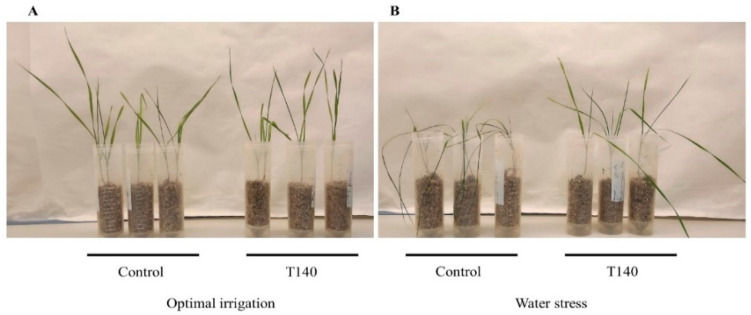
Four-week-old wheat seedlings treated with *Trichoderma asperellum* T140 or untreated (control) and subjected to two irrigation regimes: (**A**) optimal irrigation and (**B**) water stress (irrigation withdrawn during the third and fourth weeks).

**Figure 3 ijms-23-06782-f003:**
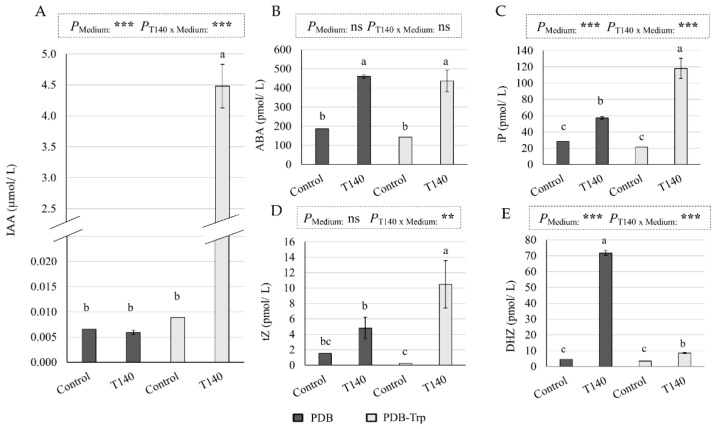
Phytohormone production by strain T140 in 4-day PDB and PDB-tryptophan (Trp) cultures compared to the basal levels of phytohormones detected in PDB and PDB-Trp control media. (**A**) Indole-3-acetic acid (IAA), (**B**) abscisic acid (ABA), (**C**) cytokinin isopentenyladenine (iP), (**D**) cytokinin trans-zeatin (tZ), and (**E**) cytokinin dihydrozeatin (DHZ) are calculated from three replicates per strain and the culture medium (*n* = 3). Values correspond to μmol or pmol per liter of culture supernatant. For each phytohormone and culture medium, different letters above the bars indicate significant differences according to one-way analysis of variance (ANOVA) followed by Tukey’s test (*p* < 0.05). For each phytohormone, a significant effect was determined using a two-way ANOVA for “T140 application”, “culture medium”, and their combination (***: *p* < 0.001; **: *p* < 0.01; ns: no statistical differences).

**Figure 4 ijms-23-06782-f004:**
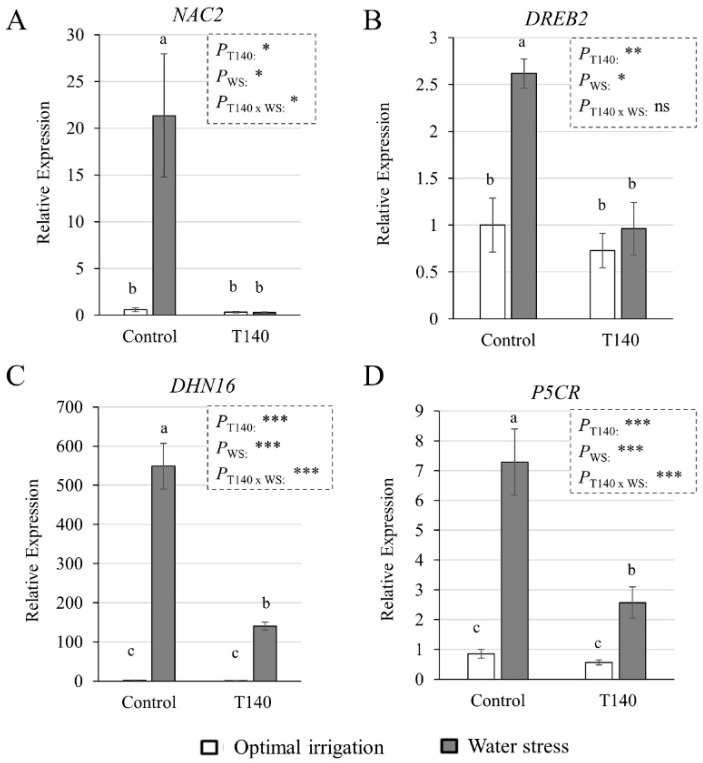
Effect of strain T140 on the expression levels of four water stress (WS) marker genes: (**A**) *NAC2*, (**B**) *DREB2*, (**C**) *DNH16*, and (**D**) P5CR, in 4-week-old wheat seedlings under optimal irrigation and WS (irrigation withdrawn during the third and fourth weeks) conditions. Data are the means of three technical replicates from three biological replicates for each treatment and irrigation condition (*n* = 3). Data are displayed as the relative quantity (RQ, 2^−ΔΔCt^) of target genes compared to the quantity of the ubiquitin gene used as a reference. Different letters above the bars indicate significant differences according to one-way analysis of variance (ANOVA) followed by Duncan’s test (*p* < 0.05). For each gene, a significant effect was determined using a two-way ANOVA for “T140 application”, “WS”, and their combination (***: *p* < 0.001; **: *p* < 0.01; *: *p* < 0.05; ns: no statistical differences).

**Figure 5 ijms-23-06782-f005:**
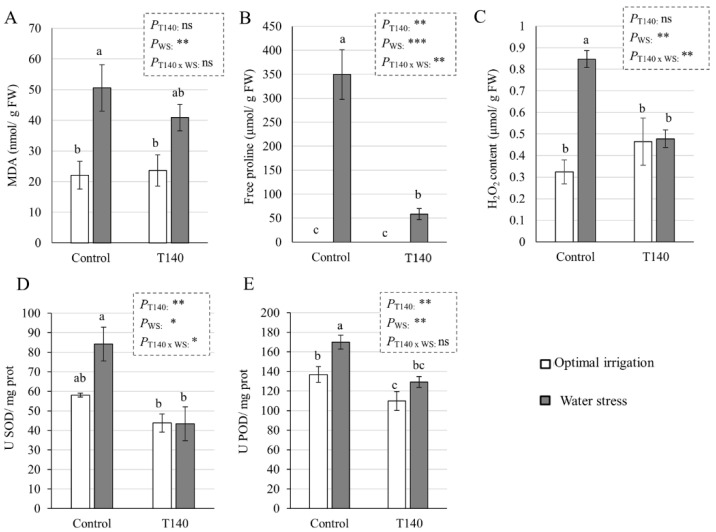
Effect of strain T140 on (**A**) MDA, (**B**) free proline, (**C**) H_2_O_2_ content, (**D**) SOD, and (**E**) POD antioxidant activities of 4-week-old wheat plants grown under optimal irrigation and water stress (WS) (irrigation withdrawn during the third and fourth weeks) conditions. Data were calculated from three replicates for each treatment and irrigation condition (*n* = 3), and different letters above the bars indicate significant differences according to one-way analysis of variance (ANOVA) followed by Tukey’s test (*p* < 0.05). For each set of data, a significant effect was determined using a two-way ANOVA for “T140 application”, “WS”, and their combination (***: *p* < 0.001; **: *p* < 0.01; *: *p* < 0.05; ns: no statistical differences). MDA: malondialdehyde, SOD: superoxide dismutase, and POD: peroxidase.

**Figure 6 ijms-23-06782-f006:**
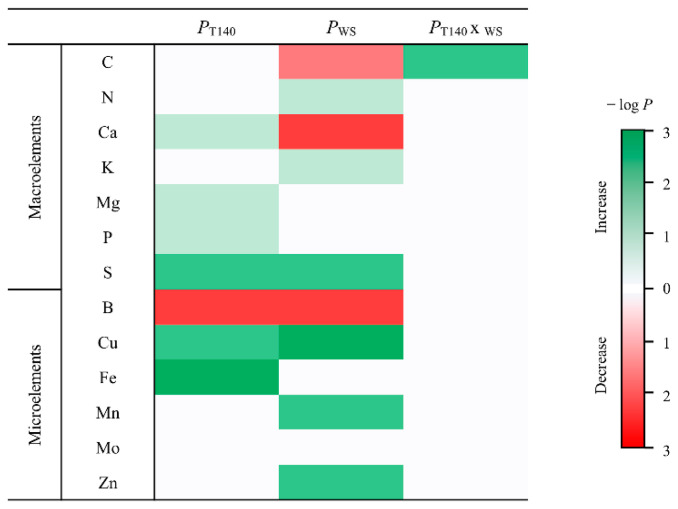
Effect of T140 application, water stress (WS, 1/2 watering applied), and the combination of both on the macro- and microelement contents of wheat grains harvested in the greenhouse assay 28 weeks after sowing. The heat map is colored according to the significance degree (–log *p*-value), with darker green indicating a higher increase and darker red indicating a greater decrease in the content. Data are calculated from five replicates per each treatment and irrigation condition (*n* = 5). For each element, a significant effect was determined by a two-way ANOVA.

**Figure 7 ijms-23-06782-f007:**
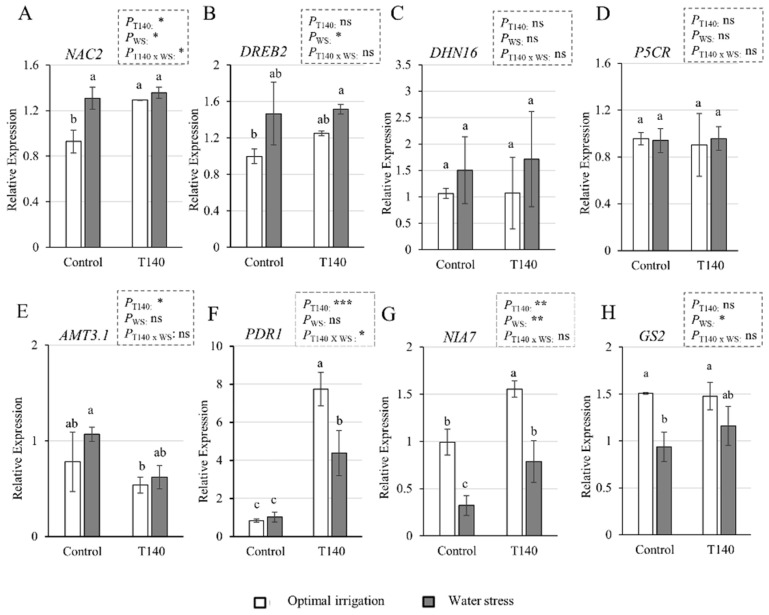
Effect of strain T140 compared to plants not treated with T140 (control) on the expression of eight genes related to drought responses or N metabolism: (**A**) *NAC2*, (**B**) *DREB2*, (**C**) *DNH16*, (**D**) P5CR, (**E**) *AMT3.1* (**F**) *PDR1*, (**G**) *NIA7*, and (**H**) *GS2* in N-fertilized 25-week-old wheat plants under optimal irrigation and water stress (1/2 watering applied) conditions. Data are the means of three technical replicates from three biological replicates for each treatment and irrigation condition (*n* = 3). Data are displayed as the relative quantity (RQ, 2^−ΔΔCt^) + standard deviations of the target genes compared to the quantity of the reference gene (ubiquitin). Different letters above the bars indicate significant differences according to one-way analysis of variance (ANOVA) followed by Duncan’s test (*p* < 0.05). For each set of data, the effect of significance was determined using a two-way ANOVA for “T140 application”, “WS”, and their combination (***: *p* < 0.001; **: *p* < 0.01; *: *p* < 0.05; ns: no statistical differences).

**Figure 8 ijms-23-06782-f008:**
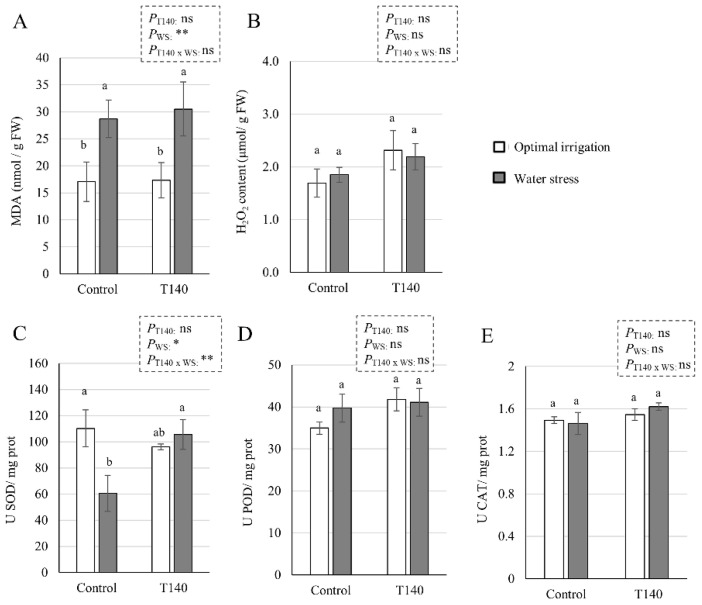
Effect of strain T140 application compared to plants not treated with T140 (control) on (**A**) the MDA and (**B**) H_2_O_2_ content, and the antioxidant activity of (**C**) SOD, (**D**) POD, and (**E**) CAT in N-fertilized 25-week-old wheat plants under optimal irrigation and water stress (WS, 1/2 watering applied) conditions. Data were calculated from three replicates for each treatment and irrigation condition (*n* = 3), and different letters above the bars indicate significant differences according to one-way analysis of variance (ANOVA) followed by Tukey’s test (*p* < 0.05). For each set of data, the effect of significance was determined using a two-way ANOVA for “T140 application”, “WS”, and their combination (**: *p* < 0.01; *: *p* < 0.05; ns: no statistical differences). MDA: malondialdehyde, SOD: superoxide dismutase, POD: peroxidase, and CAT: catalase.

**Table 1 ijms-23-06782-t001:** Effect of the application of strain T140 compared to plants not treated with T140 (control) on the dry weight of the aerial parts (DW), grain yield per plant (GYP), grain number per plant (GNP), spike number per plant (SNP), total grain yield (GY), and total spike number (SN) of highly N-fertilized 28-week-old wheat plants grown in a greenhouse under two irrigation conditions (optimal irrigation (OI), and water stress (WS, 1/2 watering applied)).

	Control		T140	
	OI	WS	OI	WS
DW (g)	9.2 ± 1.2 a	6.6 ± 1.6 c	9.1 ± 0.7 a	7.6 ± 0.9 b
GYP (g)	2.8 ± 0.4 a	2.1 ± 0.7 b	2.8 ± 0.5 a	2.6 ± 0.6 a
GNP	87.8 ± 16.2 a	63.9 ± 19.5 b	84.0 ± 13.5 a	79.7 ± 19.0 a
SNP	3.9 ± 0.8 a	3.0 ± 1.1 b	4.3 ± 0.6 a	3.7 ± 0.4 a
GY (g)	101.99	74.14	101.74	91.99
GN	3160	2300	3024	2868
SN	139	109	154	133

Data are calculated from 12 replicates per treatment and irrigation condition (*n* = 12). Values in the same row with different letters are significantly different according to one-way analysis of variance (ANOVA) followed by Tukey’s test (*p* < 0.05).

## Data Availability

Not applicable.

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
