# Peer review of "Effect of Trichoderma asperellum on Wheat Plants’ Biochemical and Molecular Responses, and Yield under Different Water Stress Conditions"

_ijms, 2022, doi:10.3390/ijms23126782_

Round 1
Reviewer 1 Report
The article “Effect of Trichoderma asperellum on wheat plants biochemical and molecular responses, and yield under different water stress conditions” presents a subject of plant research relevance and explores the literature of the area. The title and subject of the manuscript are very interesting from the methodological and practical point of view, suitable and adequate the abstract of the paper is factual concrete, realistic, understandable, self-readable. However the manuscript should be improve.
1. The introduction and objective of the study is well define and clearly demonstrate the area of the subject. However, a further improvement is required to clarify the object of the study line 85-94.
2. The results section need to be rewrite, to clearly explain the each section, as the current representation of the results is so confusing.
3. Line 103-105, the author claim that the T68 shows higher fresh weight than T49, T75, or T77 treatments under OI and T140. However, the values present in table 1 and statistical analysis shows different stories. Please preform the statistical analysis one more time to confirm.
4. Line 170 and 180, please check. Why SOD level were decreased under T140 inoculation under WS? As SOD, helps break down potentially harmful ROS molecules in cells and prevent damage to tissues.
5. Line 424, please provide the location site of the strain.
6. All the figure quality need to be improve.
Author Response
Dear Editor,
First of all we would like to thank you and the reviewers for the quick and accurate review of our article because we think that it has helped to improve it. We have sent the manuscript to MDPI English editing for checking. As we have included four new references in this new version, please note that the reference numbers have changed. As there are many small typographic changes, we have not highlighted them as they are not scientific relevant and do not answer specific questions of the reviewers.
In the supplementary material we have also made changes to the legends in Table 1 and Figures S1 and S2. In addition, we have produced a new Table S2 to adapt the reference numbers. New Tables S1 and S2 have been uploaded, and the new captions of Figures S1 and S2 pasted during the uploading process.
Here are our point-by-point responses to the comments of Reviewer 1:
- The introduction and objective of the study is well define and clearly demonstrate the area of the subject. However, a further improvement is required to clarify the object of the study line 85-94.
- We have rewritten the objective of this study and have included the following text at the end of the Introduction (new lines 143-152): “The aim of this study was (i) to select a strain of Trichoderma from a set of eight, representing different genotypes within the genus, to determine its ability to increase the tolerance of wheat plants to WS; (ii) to explore the production of phytohormones and ACCD activity related to such ability in the selected strain ( asperellum T140); (iii) to determine the extent to which T140 confers WS tolerance to wheat seedlings through the measurement of biochemical parameters and gene expression changes linked to plant responses to drought; and (iv) to validate the protective effect of T140 on wheat plants subjected to high N fertilization and moderate WS and its role in maintaining grain yield and quality parameters at the greenhouse level”.
- The results section need to be rewrite, to clearly explain the each section, as the current representation of the results is so confusing.
- A division of sub-sections has been followed for both the Results section and the Discussion. We believe it is now easier to keep track of the results.
- Results include four subsections and we have included a short introductory paragraph: 2.1. Trichoderma strain selection (new lines 155-159); 2.2. Phytohormone production by Trichoderma asperellum T140 (new lines 273-276); 2.3. Molecular and biochemical changes in wheat seedlings (new lines 297-299); and 2.4. Greenhouse trial up to grain production.
- Since ACCD activity was calculated for the eight initial Trichoderma strains considered, we have included these results at the end of the subsection 2.1. (new lines 265-271).
- Line 103-105, the author claim that the T68 shows higher fresh weight than T49, T75, or T77 treatments under OI and T140. However, the values present in table 1 and statistical analysis shows different stories. Please preform the statistical analysis one more time to confirm.
- It is true that reporting values of other Trichoderma strains may distract attention from the objective of selecting the one that performed best on plants subjected to water stress. We have deleted that sentence referring to the behavior of the plants with the unselected Trichoderma strains and included the following text (new lines 161-163): “No differences in the dry weight were detected among the treatments under either the OI or WS conditions. However, …”.
- Line 170 and 180, please check. Why SOD level were decreased under T140 inoculation under WS? As SOD, helps break down potentially harmful ROS molecules in cells and prevent damage to tissues.
- One explanation may be that water-stressed seedlings do not appear to be stressed when treated with T140. Thus, it should not be necessary to increase SOD activity as the plants do not seem to suffer from stress. In this sense, the no increase in SOD is not an isolated event and correlates with the decreased levels of H2O2 and NAC2 and DREB2 gene expression levels observed in T140-treated WS seedlings compared to controls.
- Line 424, please provide the location site of the strain.
- We have included this text in Materials and Methods (new lines 880-884): “Eight Trichoderma strains, representing different genotypes, were included in this study: parareesei T6 (air, UK) [55,56]; T. virens T49 (soil, Brazil), T. longibrachiatum T68 (soil, Brazil), T. spirale T75 (solarized soil, Spain) and T. koningii T77 (soil, Spain) [6,55]; T. harzianum T115 (soil, Philippines) [6], T. hamatum T123 (Marchantia polymorpha rhizoids, Spain), and T. asperellum T140 (strawberry nursery soil, Spain) (references of our collection, CIALE, University of Salamanca, Spain)”.
- All the figure quality need to be improve.
- The quality of all figures has been improved.
Reviewer 2 Report
Authors used in the text lots of abbreviations. It needs to bring up the explanation of these abbreviations in front of the article.
L. 433 Please explain abbreviation ACCD in the title of this section
Table 1. It is difficult to believe that aerial part of 4-week old wheat seedlings weighed only 40-100 mg. As before the Authors obtained 10-fold higher biomass in similar experiment.
Illescas, M.; Pedrero-Méndez, A.; Pitorini-Bovolini, M.; Hermosa, R.; Monte, E. Phytohormone production profiles in Trichoderma species and their relationship to wheat plant responses to water stress. Pathogens 2021, 10, 991. doi:10.3390/pathogens10080991.
Moreover dry mass should be also given.
Fig.2 The all 2-way anova results should be given (I found only medium; medium x T140 interaction – what about T140 effect?)
L. 168 not “The beneficial effect...” but “The effect of strain .....”-I would avoid such statement in this place
The content of MDA and H2O2 probably were wrong calculated/measured because the given values are extremely high (100 fold higher in relation to the other papers -https://doi.org/10.3390/plants10040733; https://doi.org/10.3390/plants9020285)
The quality of figures must be improve.
Author Response
Dear Editor,
First of all we would like to thank you and the reviewers for the quick and accurate review of our article because we think that it has helped to improve it. We have sent the manuscript to MDPI English editing for checking. As we have included four new references in this new version, please note that the reference numbers have changed. As there are many small typographic changes, we have not highlighted them as they are not scientific relevant and do not answer specific questions of the reviewers.
In the supplementary material we have also made changes to the legends in Table 1 and Figures S1 and S2. In addition, we have produced a new Table S2 to adapt the reference numbers. New Tables S1 and S2 have been uploaded, and the new captions of Figures S1 and S2 pasted during the uploading process.
Here are our point-by-point responses to the comments of Reviewer 2:
- Authors used in the text lots of abbreviations. It needs to bring up the explanation of these abbreviations in front of the article.
- A list of abbreviations has been added to the first page of the article. However, we leave it to the editor's discretion as we have not seen abbreviations summarized in this journal in a separate section.
- 433 Please explain abbreviation ACCD in the title of this section
- Abbreviations have been removed from subsection headings.
- Table 1. It is difficult to believe that aerial part of 4-week-old wheat seedlings weighed only 40-100 mg. As before the Authors obtained 10-fold higher biomass in similar experiment (Illescas, M.; Pedrero-Méndez, A.; Pitorini-Bovolini, M.; Hermosa, R.; Monte, E. Phytohormone production profiles in Trichoderma species and their relationship to wheat plant responses to water stress. Pathogens 2021, 10, 991).
- Yes, there are indeed different values, but the experiments are not comparable. In this trial, we used vermiculite alone as substrate in 50 mL tubes and in Illescas et al (2021) we grew the plants in 250 mL pot with a peat:vermiculite (3:1) commercial substrate.
- Moreover dry mass should be also given.
- We have included dry weight values in Table 1. In addition, we have included the following text (new lines 161-162): “No differences in the dry weight were detected among the treatments under either the OI or WS conditions”.
- Regarding this point, we have included the following text (new lines 505-508) in the Discussion section: “As observed by other authors [7], differences were observed in the fresh weight values between the plants with OI and WS, but this was not the case for the dry weight. Our results obtained with strain T140 are indicative of an absence of wheat plant growth promotion, regardless of WS”.
- 2 The all 2-way anova results should be given (I found only medium; medium x T140 interaction – what about T140 effect?)
- As the phytohormone production study has been performed with a single strain of Trichoderma, it does not may sense to do 2-way ANOVA to see the strain effect, as the effect of T140 compared to the controls (two different culture media) without Trichoderma can be detected with 1-way ANOVA.
- 168 not “The beneficial effect...” but “The effect of strain .....”-I would avoid such statement in this place Done
- Done (new lines 333).
- The content of MDA and H2O2 probably were wrong calculated/measured because the given values are extremely high (100 fold higher in relation to the other papers -https://doi.org/3390/plants10040733; https://doi.org/10.3390/plants9020285)
- Thank you very much for bringing this point to our attention, which we had overlooked. Concerning the measurement of the MDA content per gram of fresh plant weight, we have moved in ranges of values expressed in micro-moles as described by other authors (references 21, 24, 36, 38, 44 and 54). However, following the indications of Reviewer 2, we recalculated the MDA content, expressed in nano-moles (new Figures 4A and 7A), according to the value calculation methodology described in Todorova et al (2021) (reference 7), and the values are indeed lower.
- We have included the following text in the Discussion section (new lines 867-871): “Doubling the MDA content in drought-stressed wheat plants has also been reported in previous studies [5,7]. However, the differences in the MDA levels recorded between our plants and those of these two previous studies [5,7] may be due to the degree of the WS applied and its duration, the age of the plants and the methodology used to calculate them.
- In addition, we have also rewritten a following sentence in Materials & Methods and included a new reference [7:Todorova et al, 2021] (new lines 1053-1054): “The MDA content was calculated using 155-mM-1 cm-1 as extinction molar coefficient [7]”.
- Regarding H2O2, similarly to other authors (references 21, 38 and 54), we have moved in the range of micro-moles per gram of fresh weight of plant.
- The quality of figures must be improved.
- The quality of all figures has been improved.
Reviewer 3 Report
Dear Editors,
Thank you so much for chosing me as a reviewer of the manuscript ijms-1747317 entitled “Effect of Trichoderma asperellum on wheat plants biochemical and molecular responses, and yield under different water stress conditions”. I hope that my comments will help Authors to improve their manuscript.
Detailed remarks concerning the manuscript
1. Please give the clear purpose and scientific hypothesis of the studies together the answer to the question stated as scientific hypothesis.
2. Please give the conclusion sentence as the last sentence of the abstract.
3. I suggest to divide the “Discussion” section to the subsection per analogy to the results section.
4. The “Discussion” section should be a little bit modified as some sentences sounds like results. There are also figures citation in the “Discussion” section
Author Response
Dear Editor,
First of all we would like to thank you and the reviewers for the quick and accurate review of our article because we think that it has helped to improve it. We have sent the manuscript to MDPI English editing for checking. As we have included four new references in this new version, please note that the reference numbers have changed. As there are many small typographic changes, we have not highlighted them as they are not scientific relevant and do not answer specific questions of the reviewers.
In the supplementary material we have also made changes to the legends in Table 1 and Figures S1 and S2. In addition, we have produced a new Table S2 to adapt the reference numbers. New Tables S1 and S2 have been uploaded, and the new captions of Figures S1 and S2 pasted during the uploading process.
Here are our point-by-point responses to the comments of Reviewer 3:
- Please give the clear purpose and scientific hypothesis of the studies together the answer to the question stated as scientific hypothesis.
- As a starting point for the paper, we have included this text (new lines 145-146) at the end of the Introduction: “Trichoderma can be useful for protecting plants from drought stress, but not all strains are able to put this ability into practice”.
- Moreover, at the request of reviewer 1, we have also included the object of the work as an answer to the starting hypothesis (new lines 145-154): “The aim of this study was (i) to select a Trichoderma strain from a set of eight, representing different genotypes within the genus, by determining its ability to increase the tolerance of wheat plants to WS; (ii) to explore the production of phytohormones and ACCD activity related to such ability in the selected strain ( asperellum T140); (iii) to determine the extent to which T140 confers WS tolerance to wheat seedlings through the measurement of biochemical parameters and gene expression changes linked to plant responses to drought; and (iv) to validate the protective effect of T140 on wheat plants subjected to high N fertilization and moderate WS and its role in maintaining grain yield and quality parameters at the greenhouse level”.
- Please give the conclusion sentence as the last sentence of the abstract.
- We have included this sentence (new lines 25-27): “Results demonstrate the effectiveness of T. asperellum T140 as a biostimulant for wheat plants under WS conditions, making them more tolerant to drought”.
- In addition, we have included this sentence in the Conclusion section (new lines 1117-1119): “Since not all Trichoderma strains protect plants to the same extent from drought stress, careful selection including plant trials, and measurement of biochemical and molecular parameters in both the fungus and the plant is needed”.
- I suggest to divide the “Discussion” section to the subsection per analogy to the results section.
- As suggested by Reviewer 1, we have rewritten the sub-section headings of the Results and, following the indications of Reviewer 3, we have also divided the Discussion into four sub-sections: 3.1. Trichoderma strain selection; 3.2. Phytohormone production by T. asperellum T140; 3.3. Molecular and biochemical changes in wheat seedlings; 3.4. Greenhouse trial up to grain production.
- The “Discussion” section should be a little bit modified as some sentences sounds like results. There are also figures citation in the “Discussion” section
- Yes, the reviewer is right. We have removed any reference to tables and figures in the Discussion.
- In addition, we have rewritten some paragraphs have been rewritten:
(New lines 516-519): “As observed by other authors [7], differences were observed in the fresh weight values between the plants with OI and WS, but this was not the case for the dry weight. Our results obtained with strain T140 are indicative of an absence of wheat plant growth promotion, regardless of WS”.
(New lines 882-885): “Doubling the MDA content in drought-stressed wheat plants has also been reported in previous studies [5,7]. However, the differences in the MDA levels recorded between our plants and those of these two previous studies [5,7] may be due to the degree of the WS applied and its duration, the age of the plants and the methodology used to calculate them”.
Round 2
Reviewer 2 Report
The paper was corrected and can be published in IJMS.
Author Response
Many thanks for your positive response.